# A Positive Impact of an Observational Study on Breastfeeding Rates in Two Neonatal Intensive Care Units

**DOI:** 10.3390/nu14061145

**Published:** 2022-03-08

**Authors:** Sophie Laborie, Géraldine Abadie, Angélique Denis, Sandrine Touzet, Céline J. Fischer Fumeaux

**Affiliations:** 1Service de Réanimation Néonatale et Néonatologie, Hôpital Femme Mère Enfant, Hospices Civils de Lyon, 69677 Bron, France; 2Réanimation Pédiatrique et Médecine Néonatale, CHU Félix Guyon, 97405 Saint Denis de la Réunion, France; geraldine.abadie@gmail.com; 3Laboratoire de Biométrie et Biologie Evolutive, UMR 5558, CNRS, Université Claude Bernard Lyon 1, 69100 Villeurbanne, France; angelique.denis@chu-lyon.fr; 4Service de Biostatistique et Bioinformatique, Pôle Santé Publique, Hospices Civils de Lyon, 69003 Lyon, France; 5Laboratoire de Biométrie et Biologie Évolutive, Équipe Biostatistique-Santé, UMR 5558, CNRS, 69100 Villeurbanne, France; 6Service de Recherche Clinique et Épidémiologique, Pôle Santé Publique, Hospices Civils de Lyon, 69003 Lyon, France; sandrine.touzet@chu-lyon.fr; 7Research on Healthcare Performance Lab, Inserm U1290, Université Claude Bernard Lyon 1, 69008 Lyon, France; 8Department Mother-Woman-Child, Clinic of Neonatology, Lausanne University Hospital and University of Lausanne, 1011 Lausanne, Switzerland; Celine-Julie.Fischer@chuv.ch

**Keywords:** Hawthorne effect, neonatal intensive care unit, newborns, observer bias, breastfeeding

## Abstract

We aimed to investigate whether the participation in an observational study on breastfeeding (*Doal*) modified breastfeeding outcomes in enrolling neonatal intensive care units (NICUs). This bi-centric before-and-after study included neonates who were admitted during a 4-month period *before* and a 4-month period *after* the implementation of *Doal*. Breastfeeding intention and breastfeeding rates at discharge were compared between the two periods. The association between inclusion in *Doal* and breastfeeding at discharge was assessed among the infants fulfilling the inclusion criteria of *Doal*. The present study included 655 neonates. After adjustments, both breastfeeding (aOR 1.21, 95%CI [1.1; 1.4], *p* = 0.001) and exclusive breastfeeding (aOR 1.8, 95%CI [1.4; 2.3], *p* < 0.001) at discharge increased in the period *after*. Breastfeeding intention was higher in one center in the period *after* (79%) compared to *before* (59%, *p* = 0.019). Compared to the period *before*, neonates who were not included in *Doal* in the period *after* had a lower chance to be breastfed at discharge, whereas those included were more frequently exclusively breastfed. The participation in an observational study on breastfeeding was associated with an increase in breastfeeding outcomes in enrolling neonatal intensive care units (NICUs). Patients who are not included deserve attention as they are at risk to be disadvantaged regarding breastfeeding success.

## 1. Introduction

Mother’s own milk is the optimal milk for infant nutrition [1]. In preterm neonates, it decreases the rate of major complications [2,3,4,5] and improves long-term development [6,7,8]. Regrettably, breastfeeding rates in neonatal intensive care units (NICUs) nowadays often remain far below those observed in healthy term infants [9,10,11]. There is thus an urgent need for effective evidence-based actions enhancing breastfeeding in this vulnerable group. One of the challenges to reach this goal, at the NICU level, is to improve the behavior of both caregivers and mothers [12,13].

Participation in medical research has been increasingly recognized to modify caregivers and patients’ behavior, regardless of the study design or intervention [14]. This has been described as the “Hawthorne effect”. Mayo et al. [15] first described a change in behavior linked to the awareness of being observed which is related to a “social desirability consideration” in an industrial professional context [14]. In medical research, the Hawthorne effect can affect (when observed) behavior positively, as demonstrated in the hand hygiene compliance of caregivers [16,17,18] or in the eating patterns of young individuals [19]. Despite the preponderance of behavioral factors in the field of breastfeeding, the influence of research on breastfeeding issues remains to be investigated in NICUs.

Our hypothesis was that the participation in studies on breastfeeding issues–even observational ones–can improve breastfeeding outcomes by impacting the behavior of health workers and/or mothers. We therefore aimed herein to determine whether, and how, the implementation of an observational study on breastfeeding modified breastfeeding outcomes in the participating NICUs. 

## 2. Materials and Methods

### 2.1. Study Design, Setting, and Participants

This was a bi-centric, before-and-after study, conducted in two French university hospital NICUs, a 52-bed level 3 (center A) and an 18-bed level 2 (center B). From April 2012 until March 2014, these NICUs participated in an observational study on fresh breast milk use (the *Doal* study; see below) [20]. The current study included all the neonates admitted to one of the two participating NICUs for at least 24 h during one of the following periods: period *before*, i.e., before the implementation of the *Doal* study, from 1 October 2011 to 31 January 2012; and the period *after*, i.e., after the implementation of the *Doal* study, from 1 September 2012 to 31 December 2012.

All neonates admitted at the two NICUs for >24 h during the time periods were eligible. Neonates admitted during the period *after* were eligible for inclusion in the present study independently of their inclusion status in the *Doal* study. During the period *before*, mothers and caregivers were unaware of the *Doal* study and of the present study; during the period *after*, all caregivers had received an information on the *Doal* study (aims, outcomes, design), but were unaware of the present study. All parents of neonates included in the *Doal* study (study patients) had given informed consent; neonates not included in the *Doal* study (non-study NICU patients) were not eligible for inclusion, had parents who declined participation, or had not been approached. In the period *after*, the *Doal* study was still ongoing and no result was available.

Breastfeeding and exclusive breastfeeding were first analyzed in the overall population included. In order to take into account *Doal* inclusion status a subgroup, analysis was conducted on infants fulfilling the inclusion criteria of the *Doal* study (i.e., hospitalized before 7 days of life, mother intending to breastfeed).

### 2.2. The Doal Study

The *Doal* study was a prospective, observational, cohort study aimed at reporting the use of raw mother’s own milk in the study NICUs among mothers intending to breastfeed. It also aimed to investigate if the early administration of raw mother’s own milk (before 7 days of life) was associated with breastfeeding continuation [20]. Mothers of infants hospitalized before 7 days of life and at least during 24 h were eligible, the study was then presented as often as possible, and informed consent was necessary for inclusion. At 7 days of life, the included mothers were interviewed to collect data on factors known to be related to breastfeeding such as socioeconomic status, previous experience of breastfeeding, antenatal breastfeeding information, milk volume monitoring, and breastfeeding support. Breastfeeding continuation was assessed at discharge according to the medical chart, and at 2 and 6 months of corrected age by a phone call. Nurses or doctors were not interviewed, but the frequency of any direct breastfeeding during hospitalization and the delay before first raw milk administration, first direct breastfeeding, first oral feeding, and first skin-to-skin contact were systematically extracted for the included infants. During the *Doal* study period, 47% of all hospitalized infants were enrolled in the *Doal* study.

### 2.3. Breastfeeding Policies in the Participating NICUs

In the participating NICUs, mother’s own milk was considered as the reference milk. Breastfeeding counter-indications were rare and included galactosemia, maternal HIV infection, indispensable maternal treatment not compatible with breastfeeding, and untreated addiction to illicit drugs. When direct breastfeeding was not possible, mothers intending to breastfeed were encouraged to express their milk as soon as possible after birth. When enough maternal milk was not available, maternal milk was completed with pasteurized human donor milk until infants reached 1.5 kg and 32 weeks of corrected age; thereafter they received preterm formula until 3 kg and subsequently term formula. There was no change in the policies between the two periods. All nurses had received basic training to help mothers to initiate and sustain lactation and their number did not change during the study. A lactation consultant was available in each NICU, assisted by a group of referent nurses involved in reinforcing breastfeeding support in the two units. No change in these teams occurred during the present study. 

### 2.4. Breastfeeding Outcomes

The primary outcome of this before-and-after was breastfeeding at discharge, defined as feeding with any mother’s own milk during the 48 h before hospital discharge. Other outcomes were exclusive breastfeeding at discharge, defined as exclusive feeding of mother’s milk during the 48 h before hospital discharge, and breastfeeding intention, defined as maternal intention to breastfeed their offspring at the NICU admission. Caregivers routinely collected the latter information at the first contact with the family in the NICU.

### 2.5. Data Collection

Breastfeeding outcomes, maternal characteristics (age, parity, smoking status, multiple birth, intention to breastfeed), and neonatal characteristics (gestational age, birth weight, sex, center, length of stay, and outcome at discharge) were extracted from patients’ hospital electronic medical files (IntelliSpace Critical Care an Anesthesia version H.02; Philips N.V., Koninklijke, The Netherlands). Breastfeeding outcomes were also verified manually using patient records.

### 2.6. Statistical Analysis

Assuming a breastfeeding rate at discharge of 60%, 712 neonates were expected to be necessary to detect a 10% increase in breastfeeding between the period *before* and the period *after*, with an 80% power and an alpha risk of 5%. After comparing newborns admitted during the period *before* to those admitted during the period *after* in the total population, a subgroup analysis was conducted. The latter aimed to compare neonates meeting the *Doal* study inclusion criteria (i.e., admitted before day 7 of life, staying >24 h and whose mothers intended to breastfeed) during the period *after* to those meeting the same criteria during the period *before*.

The association between study period and breastfeeding or exclusive breastfeeding at discharge was assessed using logistic regression, controlling for maternal and neonatal factors significantly associated with the outcome. The variables tested were NICU, maternal age, smoking, parity, sex, gestational age, birth weight, multiple gestation, length of stay, and outcome at discharge.

The model was fitted using a General Estimation Equation method to account for neonates nested within centers. A stratified analysis by center assessed the association between periods and breastfeeding intention. Results were expressed as adjusted odds ratios (OR) and their 95% confidence intervals (95%CI).

The breastfeeding intention was compared between study period using chi-square and a stratified analysis by center to control for confounders. Analyses were performed using SAS version 9.3 software (SAS Institute, Cary, NC, USA).

### 2.7. Ethics

The present study (“Effets d’une étude observationnelle sur les dons directs de lait maternel, et sur le taux d’allaitement des nouveaux-nés hospitalisés”) was approved by the institutional review board (Comité d’Ethique du CHU de Lyon) on 29 May 2013.

## 3. Results

### 3.1. Population

During the two periods, a total of 655/658 admitted neonates were included in the overall analysis, and 422 in the subgroup analysis (Figure 1). The maternal and neonatal characteristics of included infants in the two periods were comparable (Table 1).

### 3.2. Breastfeeding Outcomes in the Total Population According to the Period

#### 3.2.1. Any Breastfeeding

At hospital discharge, 181/301 (60%) infants were breastfed in the period *before*, and 215/354 (61%) in the period *after*. Several factors were significantly associated with any breastfeeding at discharge in univariate analysis (Table 2).

After adjustment, the rate of any breastfeeding at discharge was significantly higher in the period *after* compared to the period *before* (aOR 1.21, 95%CI (1.08; 1.36), *p* = 0.0013). In this multivariable model, only gestational age (aOR for 1 week increase 1.12, 95%CI (1.09; 1.14), *p* < 0.0001) and smoking (aOR 0.45, 95%CI (0.43; 0.47), *p* < 0.0001) were also significantly associated with breastfeeding at discharge. 

#### 3.2.2. Exclusive Breastfeeding

At hospital discharge, 36/301 (12%) neonates were exclusively breastfed in the period *before* and 64/354 (18%) in the period *after*. Results of the univariate analysis are presented in Table 2. After adjustment, exclusive breastfeeding was significantly higher during the period *after* compared to the period *before* (aOR 1.76, 95%CI (1.36; 2.28), *p* < 0.001). In this multivariable analysis, the only other variable that was significantly associated with exclusive breastfeeding was multiple birth (aOR 0.45, 95%CI (0.32; 0.63) *p* < 0.001).

#### 3.2.3. Breastfeeding Intention

Breastfeeding intention was not significantly different between the period *before* and the period *after* (70% versus 75%, *p* = 0.674) in the total population. In Center B breastfeeding intention was more frequent in the period *after* (79%) than in period *before* 59%, *p* = 0.019), and remained similar in Center A (73% versus 74%, *p* = 0.401, respectively).

### 3.3. Breastfeeding Outcomes According to Doal Inclusion Status

This analysis included 185 infants from the period *before* and 237 from the period *after* who were admitted before day 7 of life and intended to be breastfed (Figure 1). The characteristics of these neonates are presented in Table 3. The sub-groups were significantly different in terms of gestational age, birth weight, multiple gestation, outcome at discharge, and length of stay.

The results of the univariate analysis of breastfeeding and exclusive breastfeeding are presented on Table 4.

In multivariable analysis, non-study NICU patients in the period *after* had a significantly lower probability of being breastfed (any breastfeeding) at discharge as compared to neonates of the period *before* (aOR 0.61, 95%CI (0.40; 0.92), *p* = 0.0193), and inclusion in the *Doal* study nearly tripled the chances of being exclusively breastfed (aOR 2.84, 95%CI (2.16; 3.73); *p* < 0.0001, Table 5). The other significant variables are presented in Table 5.

## 4. Discussion

Herein, comparison of breastfeeding outcomes before and after the implementation of a large observational study on breastfeeding in two NICUs found a significant improvement in both any breastfeeding and exclusive breastfeeding at discharge in the latter period. During the period *after*, study patients had a greater chance of being exclusively breastfed than in the period *before*. However, during the period *after*, non-study NICU patients had a lower chance of being breastfed than in the period *before*. In addition, in one of the two participating centers (Center B), breastfeeding intention was significantly more frequent in the period *after* than in the period *before*.

The discrepancy between intention of breastfeeding and breastfeeding rates at discharge during both periods suggested that there was room for improvement in breastfeeding support in both NICUs, and the factors associated with this have been explored in the *Doal* study [20]. The rise in the overall breastfeeding rates in the period *after*, regardless of being included in the study or not, could be related to, among other aspects, improvement of caregiver behavior, technical knowledge, and their awareness of the importance given to breastfeeding. Furthermore, the important positive impact of being included in the study also suggests a possible motivational reinforcement in the enrolled mothers. Thus, the Hawthorne effect may play a key role both in caregivers and patients’ behavior. According to Coombs et al., the Hawthorne effect “provides a confirmation of how researchers may successfully interact within a social context to bring about positive changes in both attitudes and task performances” [21]. In the medical field, the Hawthorne effect has been particularly well recognized in hand hygiene [16,17,18,22]. Although it has since been reported to drive physiological changes with improvement in some laboratory variables, such as in obese patients [23], the Hawthorne effect remains best established in changing practices and behaviors. In neonatology, it has mainly been shown to reduce the rate of medical errors in neonatal care units [24]. This study is the first, to our knowledge, to report an impact on breastfeeding. 

In addition to the Hawthorne effect, other factors could have contributed to the observed differences: enrolled mothers might have received additional support through further attention or counseling from the investigators during their enrollment and the interview. Furthermore, a selection bias of mothers enrolled in the *Doal* study could also contribute to this as suggested by the discrepancy between length of stay in study patients and non-study NICU patients; multivariable analysis was adjusted on length of stay for any breastfeeding and this was not significant for exclusive breastfeeding. However, we can also speculate that mothers who were more present at the bedside of their infants, more fluent in French, or more participative could be both more susceptible to be included in the *Doal* study and more likely to successfully breastfeed.

Results did differ between the two centers as Center B had significantly lower breastfeeding intention at baseline and this increased significantly between the periods whereas for Center A no significant increase was found. Mothers whose fetus were at risk of hospitalization routinely encounter a pediatrician before birth; therefore, the importance given to breastfeeding during this meeting may have been enhanced during the period *after* in Center B. Such differential effects have been reported for hand hygiene, although published studies are not concordant as a greater Hawthorne effect has been reported in units with lower baseline performance [25], but also in units with higher baseline performance [26].

Conversely, breastfeeding success decreased in non-study NICU patients after *Doal* implementation. This fall was not observed in a hand hygiene study in which baseline level and unobserved and observed level of compliance were available with an automatic measure [16,17]. However, similar unexpected undesirable consequences of studies have been described in quality improvement studies [27,28]. This finding herein could be related to the multiple roles that nurses have in NICUs; they may have provided a greater effort to support breastfeeding for study patients and consequently impaired the support of those not included.

One strength of the present study is that it concerned almost all infants hospitalized in both neonatal units, composed of a mixed population of extreme preterm, preterm, and term infants, with various diseases or malformations. Due to the recruitment of the NICUs, the number of extremely preterm infants was limited. One limitation is that we could not adjust all variables known to be related to breastfeeding, such as the socioeconomic characteristics of the mothers or previous experience of breastfeeding, because such information was lacking for the non-study NICU patients. Thus, we cannot exclude that a part of the observed effect is due to uncontrolled differences and biases, and not only to the Hawthorne effect as described by Rosenberg et al. in a study about school enrollment and HIV prevention [29]. Another limitation is that the long-term impact on breastfeeding outcomes was not investigated. Additional research is required to determine whether the improvement in breastfeeding outcomes would be sustained beyond the study period: after an initial peak, the Hawthorne effect is known to diminish owing to habituation [14,25], and in hand hygiene studies its effect did not exceed the duration of observation [30,31]. Furthermore, the generalization of the results may also be affected by variations in the Hawthorne effect between units [26,32].

Despite its limitations, the results constitute novel information from both a clinical point of view, as it may constitute a potential strategy to improve breastfeeding rates in the NICUs, and from a methodological point of view, as it should be recognized that an observational study on breastfeeding might influence the measured outcome, with a lack of reproducibility between centers. In any case, these findings advocate for the development of additional research in this area.

## 5. Conclusions

The implementation of a large observational study on breastfeeding in two NICUs was associated with an increase in breastfeeding and exclusive breastfeeding rates at discharge. Special attention should be given to patients who are not included in clinical studies as breastfeeding may decrease among these patients.

## Figures and Tables

**Figure 1 nutrients-14-01145-f001:**
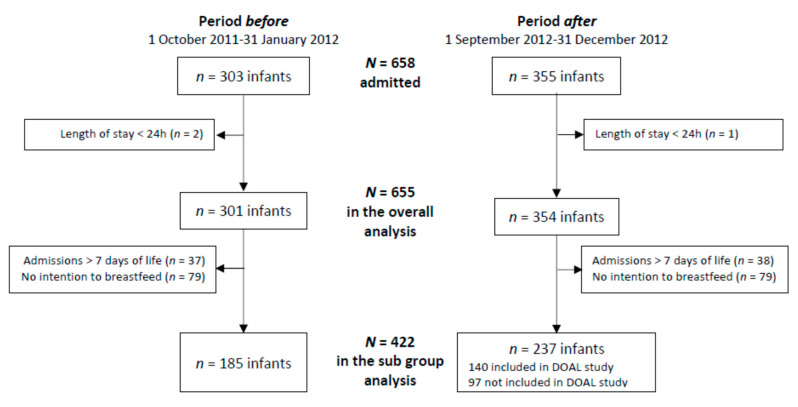
Study flow chart.

**Table 1 nutrients-14-01145-t001:** Perinatal characteristics of the population *before* and *after* the implementation of the *Doal* study.

Characteristics	Period *Before* *n* = 301	Period *After* *n* = 354	*p*-Value
NICU, *n* (%)			0.061
Center A	236 (78.4)	255 (72.0)	
Center B	65 (21.6)	99 (28.0)	
Maternal age (years), mean (SD)	30.5 (5.5)	30.8 (5.4)	0.498
Parity, mean (SD)	2.1 (1.1)	2.2 (1.1)	0.195
Smoking, *n* (%)			0.212
No	153 (50.8)	223 (63.0)	
Yes	53 (17.6)	59 (16.7)	
Missing data	95 (31.6)	72 (20.3)	
Male, *n* (%)	160 (53.2)	199 (56.2)	0.433
Gestational age (weeks), median (range)	34.6 (24.1–42)	34.6 (25.6–42.1)	0.808
Birth weight (g), median (range)	2050 (490–4320)	2100 (530–5270)	0.401
Multiple births, *n* (%)	65 (21.6)	96 (27.1)	0.102
Outcome at discharge, *n* (%)			0.851
Home	175 (58.1)	202 (57.1)	
Other hospital	118 (39.2)	140 (39.5)	
Death	8 (2.7)	12 (3.4)	
Length of stay (days), median (range)	13 (1–144)	14 (1–143)	0.729

SD: standard deviation, *p*-values were calculated using student’s or Wilcoxon’s *t*-test: Wilcoxon for quantitative variables or Chi² or Fisher’s exact test for qualitative variables.

**Table 2 nutrients-14-01145-t002:** Perinatal characteristics associated with breastfeeding and exclusive breastfeeding at discharge (univariate analysis).

Characteristics	Any Breastfeeding		Exclusive Breastfeeding	
	OR_crude_ [95%CI]	*p*-Value	OR_crude_ [95%CI]	*p*-Value
Period *after* (ref: period *before*)	1.03 [0.75; 1.40]	0.8753	1.62 [1.05; 2.52]	0.0310
NICU (ref: center A)	1.11 [0.78; 1.59]	0.5611	0.52 [0.30; 0.92]	0.0253
Maternal age (+1 year)	1.01 [0.98; 1.04]	0.5817	1.00 [0.96; 1.04]	0.8865
Smoking (ref: no)	0.47 [0.31; 0.72]	0.0005	0.87 [0.49; 1.56]	0.6456
Parity (+1 infant)	0.92 [0.80; 1.05]	0.2158	0.83 [0.68; 1.03]	0.0871
Sex (ref: male)	1.03 [0.75; 1.41]	0.8668	1.04 [0.68; 1.59]	0.8598
Gestational age (+1 week)	1.08 [1.03; 1.12]	0.0003	1.00 [0.95; 1.06]	0.9577
Birth weight (+100 g)	1.03 [1.01; 1.05]	0.0028	1.00 [0.97; 1.02]	0.7773
Multiple gestation (ref: no)	1.14 [0.79; 1.64]	0.4968	0.49 [0.28; 0.88]	0.0174
Length of stay (+7 days)	0.96 [0.91; 1.00]	0.0552	1.06 [1.00; 1.12]	0.0498
Outcome at discharge (ref: Home):				
Other hospital	1.02 [0.74; 1.41]	0.8995	0.52 [0.32; 0.82]	0.0056
Death	0.53 [0.21; 1.31]	0.1675	0 [0; infinity]	0.9996

*p*-values were calculated using the Wald test in a logistic regression.

**Table 3 nutrients-14-01145-t003:** Perinatal characteristics of neonates admitted before day 7 of life and intended to be breastfed during the period *before* and during the period *after* according to their inclusion in the *Doal* study.

	Period *Before*	Period *After*		
Characteristics	*n* = 185	Study Patients *n* = 140	Non-Study NICU Patients *n* = 97	*p*-Value
NICU, *n* (%)				0.086
Center A	154 (83.2)	104 (74.3)	72 (74.2)	
Center B	31 (16.8)	36 (25.7)	25 (25.8)	
Maternal age (years), mean (SD)	30 (5.1)	31.2 (5.5)	31 (5)	0.104
Parity, mean (SD)	2 (1.1)	2.3 (1.1)	2.1 (1.2)	0.057
Smoking, *n* (%)	26 (14.1)	21 (15)	15 (15.5)	0.420
Male, *n* (%)	104 (56.2)	77 (55)	49 (50.5)	0.652
Gestational age (weeks), median (range)	35.0 (25.3–41.3)	33.8 (27.0–42.1)	37.0 (26.9–41.9)	<10^−3^
Birth weight (g), median (range)	2100 (514–4320)	1935 (594–5270)	2800 (685–4300)	<10^−3^
Multiple birth, *n* (%)	41 (22.2)	51 (36.4)	12 (12.4)	<10^−3^
Outcome at discharge, *n* (%)				<10^−3^
Home	113 (61.1)	111 (79.3)	27 (27.8)	
Other hospital	70 (37.8)	29 (20.7)	64 (66)	
Death	2 (1.1)	0 (0)	6 (6.2)	
Length of stay (days), median (range)	14 (1–144)	31 (2–102)	5 (1–127)	<10^−3^

SD: standard deviation; *p*-values were calculated using ANOVA or Kruskall Wallis test for quantitative variables or using Chi² or Fisher’s exact test for qualitative variables.

**Table 4 nutrients-14-01145-t004:** Univariate analysis of any breastfeeding and exclusive breastfeeding among infants according to *Doal* inclusion status.

	Any Breastfeeding		Exclusive Breastfeeding	
Characteristics	OR_crude_ [95%CI]	*p*-Value	OR_crude_ [95%CI]	*p*-Value
Study period (ref *before*)	1.02 [0.75; 1.40]	0.8966	1.64 [1.00; 2.67]	0.0496
Study group (ref *before*)		0.0560 ^†^		0.0041 ^†^
*After* included	0.55 [0.3; 0.99]	0.0444	2.21 [1.30; 3.77]	0.0034
*After* not included	1.17 [0.54; 2.49]	0.6930	0.95 [0.48; 1.86]	0.8698
Maternal age (+1 year)	1.01 [0.96; 1.06]	0.7198	1.00 [0.95; 1.04]	0.9445
Smoking (ref No)	0.64 [0.32; 1.28]	0.2036	1.13 [0.59; 2.16]	0.7110
Parity (+1 child)	0.95 [0.76; 1.19]	0.6559	0.87 [0.70; 1.08]	0.2176
Sex (ref male)	0.9 [0.53; 1.53]	0.6993	1.02 [0.64; 1.64]	0.9197
Gestational age (+1 week)	1.25 [1.16; 1.35]	<0.0001	1.02 [0.96; 1.08]	0.5326
Birth weight (+100 g)	1.1 [1.06; 1.14]	<0.0001	1.00 [0.98; 1.03]	0.8789
Multiple birth	0.83 [0.46; 1.5]	0.5357	0.42 [0.22; 0.81]	0.0099
Length of stay (+7 days)	0.81 [0.76; 0.87]	<0.0001	1.03 [0.96; 1.09]	0.4152
Outcome at discharge (ref: home)		0.0144 ^†^		0.0433
Other hospital	2.33 [1.26; 4.32]	0.0071	0.52 [0.31; 0.87]	0.0122
Death	0.71 [0.14; 3.62]	0.6798	0	0.9997
Center (reference A)	0.88 [0.46; 1.7]	0.7024	0.45 [0.23; 0.90]	0.0228

^†^ global *p*-value.

**Table 5 nutrients-14-01145-t005:** Multivariable logistic regression model predicting breastfeeding at hospital discharge in neonates admitted before day 7 of life and intended to be breastfed.

Characteristics	Any Breastfeeding		Exclusive Breastfeeding	
	OR_adjusted_ [95% CI]	*p*-Value *	OR_adjusted_ [95% CI]	*p*-Value
Period (ref: period *before*)		0.2073		<0.0001
Period *after* not included	0.61 [0.40; 0.92]	0.0193	0.90 [0.59; 1.39]	0.6438
Period *after* included	0.57 [0.24; 1.36]	0.2073	2.84 [2.16; 3.73]	<0.0001
Sex (ref: male)	1.13 [1.01; 1.26]	0.0302		
Multiple birth	1.42 [0.90; 2.24]	0.1349	0.30 [0.29; 0.30]	<0.0001
Smoking	0.42 [0.40; 0.44]	<0.0001		
Gestational age (+1 week)	1.18 [1.11; 1.25]	<0.0001		
Length of stay (+7 days)	0.90 [0.89; 0.92]	<0.0001		
Outcome at discharge (ref: Home)		0.0010		
Other health structure	2.18 [1.37; 3.47]	0.0010		
Death	0.60 [0.53; 0.68]	<0.0001		

* Logistic model using GEE estimation to account for children clustered within centers.

## Data Availability

The data presented in this study are available on request from the corresponding author. The data are not publicly available due to local policy.

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
