# Peer review of "A Positive Impact of an Observational Study on Breastfeeding Rates in Two Neonatal Intensive Care Units"

_nutrients, 2022, doi:10.3390/nu14061145_

Round 1
Reviewer 1 Report
In this paper, Laborie et al have examined data from a previously conducted (and published) study to determine whether participation in an observational study on breastfeeding modified breastfeeding outcomes in participating NICUs. The authors find that both breastfeeding and exclusive breastfeeding at discharge increased in the period following the study in patients included in the study and that participation in an observational study on breastfeeding was associated with an increase in breastfeeding. This paper explores the presence and significance of the Hawthorne effect in this study population and provides an interesting insight. The data tables are well presented. The only suggestion for revision is that the authors might clarify the way the results are presented and discussed in the text. It is confusing to follow their interpretation of the results – they might use shorter phrases such as “study infants” to identify infants included in the observational study relative to “non-study NICU patients”. No other issues noted.Author Response

Reviewer 2 Report
This is a well-written clinical article on the effect of participation in an observational neonatal feeding study on breastfeeding outcomes in two French NICUs. The researchers were able to demonstrate that participation in a breastfeeding observation study already resulted in a higher rate of breast milk feeding in the NICU after the study period than prior to the study.
This is an exciting and innovative approach and shows that even the preoccupation with the topic of breastfeeding can change behavior. The authors also compare this behavioral effect, which has been described in the literature as the "Hawthorne effect", with thematically different study designs.
Therefore, this study is important to show how breastfeeding in the neonatal intensive care unit (NICU) can be made more visible to parents and staff, even without direct intervention, in order to improve the low breastfeeding rates in the NICU.
The following aspects might be added:
Material
The authors describe that this evaluation was performed before and after the observational breastfeeding study (Doal); chapter 2.2 briefly lists the Doal study and cites the previously published study (Fischer Fumeaux, C.J., Neonatology 2018, 113, 131-139).
However, only the abstract of this article is freely available, the name "Doal study" is not found in the abstract. Unfortunately, it is not entirely clear what exactly was done in the Doal study. It would be very helpful if it were described how the Doal study was conducted. What and how often were the parents interviewed? Were questionnaires also filled out by the nurses and doctors? Were the results of the study known to the nurses/parents/doctors during the “after Doal study” period?
It is also not clear to me whether the parents were asked about their breastfeeding intentions in the period before and after the Doal study before/after admission to the NICU, or whether the parents did not have to fill in any questionnaires about feeding at all.
Results
- The authors investigated whether the newborns were partially or exclusively fed with mother's own milk at discharge. Are there also data on breastfeeding, i.e. breastfeeding at the breast? Are there data on the frequency of attempts at latching on?
- The numbers in Table 3 should all be rounded equally (i.e. not 35 vs. 33.8 weeks or lenght of stay 14 vs. 33.5 days).
Discussion:
- Intention to breastfeed ranged from 59% to 79% before and after the Doal study. The discussion could briefly address the discrepancy with the rates of 60 and 61 % for "any breastfeeding" achieved at discharge. Why are they lower than previously desired by mothers? Shouldn't breastfeeding in the NICU be even better supported because of its various benefits for the newborns - and how?
- The length of NICU stay was significantly longer in the period after the Doal study (14 vs 32 days)- what explanation is there for this (Table 3)? What influence could this have on the results?
- The mean gestational age of the patients in the neonatal intensive care unit was remarkably high (35 and 34 weeks respectively) - I would have expected a higher proportion of very small preterm infants- how can this be explained? Especially for the inclusion of small preterm infants, I found the inclusion period of 4 months (before and after the Doal study for this analysis) very short, because the length of stay for small preterm infants alone is 3-4 months, so only a few of these children can be included. Why was the period not longer than 4 months?
